# Human Beta Defensin-2 mRNA and Proteasome Subunit β Type 8 mRNA Analysis, Useful in Differentiating Skin Biopsies from Atopic Dermatitis and Psoriasis Vulgaris Patients

**DOI:** 10.3390/ijms25179192

**Published:** 2024-08-24

**Authors:** Agnieszka Terlikowska-Brzósko, Ryszard Galus, Piotr Murawski, Justyna Niderla-Bielińska, Izabela Młynarczuk-Biały, Elwira Paluchowska, Witold Owczarek

**Affiliations:** 1Department of Dermatology, Military Institute of Medicine—National Research Institute, 04-141 Warsaw, Poland; epaluchowska@wim.mil.pl (E.P.); wowczarek@wim.mil.pl (W.O.); 2Department of Histology and Embryology, Faculty of Medicine, Medical University of Warsaw, 02-004 Warsaw, Poland; ryszard.galus@wum.edu.pl (R.G.); jniderla@wum.edu.pl (J.N.-B.); izabela.mlynarczuk-bialy@wum.edu.pl (I.M.-B.); 3Information and Communication Technology Department, Military Institute of Medicine—National Research Institute, 04-141 Warsaw, Poland; pmurawski@wim.mil.pl

**Keywords:** atopic dermatitis (AD), human beta defensin-2 (hBD-2), involucrin (IVL), proteasome subunit beta type 8 (PSMB8), psoriasis vulgaris (PV), tripeptidyl peptidase 2 (TPP2)

## Abstract

(1): Atopic dermatitis and psoriasis vulgaris are chronic, inflammatory diseases. Clinical presentation usually leads to a proper diagnosis, but sometimes neither clinical examination nor histopathological evaluation can be conclusive. Therefore, we aimed to build up a novel diagnostic tool and check it for accuracy. The main objective of our work was to differentiate between healthy skin (C), atopic dermatitis (AD) and psoriasis vulgaris (PV) biopsies on the base of involucrin (IVL) and human β-defensin-2 (hBD-2) concentrations and their mRNA, as well as mRNA for TPP2 and PSMB8. (2): ELISA for IVL and hBD-2 proteins and Real-time PCR for the relative expression of mRNA for: IVL (IVL mRNA), hBD-2 (hBD-2 mRNA), PSMB8 (PSMB8 mRNA) and TPP2 (TPP2 mRNA), isolated from skin biopsies taken from AD and PV patients and healthy volunteers were performed. (3): hBD-2 mRNA and PSMB8 mRNA correlated with some parameters of clinical assessment of inflammatory disease severity. hBD-2 mRNA expression, exclusively, was sufficient to distinguish inflammatory skin biopsies from the healthy ones. (4): hBD-2 mRNA and PSMB8 mRNA analysis were the most valuable parameters in differentiating AD and PV biopsies.

## 1. Introduction

Atopic dermatitis (AD) and psoriasis vulgaris (PV) are chronic inflammatory skin diseases diagnosed on the base of clinical symptoms. The frequency of AD is around 30% among children and between 2 and 17.6% in adults [1]. The disease is characterized by pruritic eczematous lesions that are often localized on the face and flexural parts of extremities. Primary lesions are exudative papules that are accompanied by excoriations and lichenification as a result of recurrent scratching [2,3]. A histopathologic picture reveals acute, subacute or chronic inflammation with varying degrees of spongiosis [4]. Total IgE immunoglobulin level may be elevated. Some AD patients suffer from asthma, allergic rhinitis or allergic conjunctivitis [5]. The diagnosis is mostly based on the clinical criteria described by Hanifine and Rajka or Wiliams [6,7,8]. The disease severity is usually assessed using the EASI (Eczema Area and Severity Index), SCORAD (SCORing Atopic Dermatitis) and IGA (Investigator Global Assessment) scales [7].

The prevalence of psoriasis in adults is between 0.14% and 1.99%, while in children, it is between 0.02% and 0.22% [9,10,11,12]. The disease usually affects non-hairy skin, although it may involve the scalp, nails, mucous membranes and joints. Primary lesions of PV are dermo-epidermal papules covered with silvery white scales [13]. There are two subtypes of skin psoriasis: psoriasis vulgaris and pustular psoriasis. Both may evolve into erythroderma. Depending on the activity of the disease, psoriasis vulgaris may be active guttate, unstable exudative or chronic stable. Lesions can be scattered on the trunk, head and extremities or localized. Although inverted psoriasis has a predilection to skin folds, psoriatic lesions are usually found on the extensor surfaces, unlike most of the AD changes. Primary lesions of PV are dermo-epidermal papules covered with silvery white scales [13]. A histopathological examination of a fully developed psoriatic papule reveals hyperkeratosis with parakeratosis and clusters of neutrophils in stratum corneum, accompanied by hypogranulosis and acanthosis with elongated rete ridges and drawn out, widened blood vessels [14,15]. In most cases, psoriasis vulgaris can be easily recognized upon clinical and histopathological examinations. Its severity can be assessed by PASI (Psoriasis Area Severity Index) and PGA (Physician’s Global Assessment) scores [13,16,17].

Clinically, AD may mimic PV; moreover, primary diagnosed AD may transform into PV and vice versa. There is an observation of coexistence of PV and AD features in the same inflammatory lesions, especially in children. These lesions are called PD (psoriasis dermatitis) or PsEma, and with time can evolve into either AD or PV [18]. Additionally, a histopathological examination may show both spongiosis and parakeratosis, with hypogranulosis that makes differential diagnosis impossible [19]. There is no commonly available and clinically approved test for differentiation between biopsies from AD and PV lesions.

Recent studies draw attention to the imbalance in proteolytic systems, resulting in an increased presentation of autoantigens that may participate in the pathogenesis of AD and PV [20,21,22]. The main cellular cytosolic proteolytic system involves the proteasome and the immunoproteasome, and since the proteasome generates oligopeptides, behind it, a set of other peptidases are recruited, including tripeptidyl peptidase 2 (TPP2) [23,24,25,26]. Proteasome subunit β type 8 (PSMB8) is a catalytic subunit of an immunoproteasome [27]. This subunit is classified as iβ5 and is responsible for the chymotrypsin-like activity of the immunoproteasome, participating in the generation of antigens to the immune system [23,28]. Up to now, the post-transcriptional upregulation of constitutive proteasomal activity was reported in psoriatic skin [29]. Tripeptidyl-peptidase 2 (TPP2) is a huge cytosol exopeptidase, and together with proteasome it plays a role in protein degradation [30] and in antigen processing, including autoantigen production [26,31,32]. It was reported that humans with homozygous *TPP2* deficiency are predisposed to develop autoimmune disorders, including atopic dermatitis [33]. Dermatitis associated with AD and PV correlated with altered proteasome expression [29], and *TPP2* deletion promoted AD [33].

Innate and acquired immunity abnormalities, as well as epidermal barrier dysfunction, have been described in both AD and PV [34,35]. The most superficial part of the epidermis—stratum corneum, consisting of corneocytes—builds the mechanical skin barrier. The inner skeleton of each corneocyte is composed of a network of proteins. Involucrin (IVL), produced by keratinocytes, is cross-linked with other proteins beneath the cell membranes of corneocytes by transaminases [36]. IVL expression is higher in the psoriatic compared with the healthy skin epidermis [37].

Human β-defensin-2 (hBD-2) is an antimicrobial peptide strengthening the chemical and immunological skin barrier. This small cationic protein eliminates some bacteria and yeasts. It also has chemotactic activity towards T lymphocytes, dendritic cells and mast cells [38]. For the first time, it was isolated from psoriatic scales [39].

According to the literature [34,35,36,37,38,39,40,41], involucrin and human β-defensin-2 seemed to be good candidates for differentiation between atopic dermatitis and psoriasis vulgaris, as their higher concentrations were described in psoriasis vulgaris. Neither TPP2 nor PSMB8 are recognized biomarkers in differentiating the diagnosis of AD and PV. However, since their participation in connection with these diseases has been described in a few works [29,33], we decided to investigate whether these two molecules differ between AD, PV and controls. We decided to rely on the mRNA expression analysis of all proteins to be able to compare their expression at the mRNA level. Involucrin is a major marker of skin cells differentiation [37,42], and human beta defensin-2 is a marker of inflammation [38,43,44]. In turn, for the immunoproteasome, the activity—the essential subunit is iB5 (PSMB8) [22,28,45]—and for TPP2—the most conserved protease—we chose the most frequently studied sequence to have a reference for other works [30,46]. Since the size of the sample that can be taken from the patient is limited, it was more reasonable to study the reference molecules at the mRNA level. Additionally, for markers of inflammation and skin differentiation, we also studied expression at the protein level to have a second point of reference.

The aim of our work was to verify if it is possible to differentiate tissues from AD, PV and C participants based on analysis of the six parameters: IV, hBD-2, IVL mRNA, hBD-2 mRNA, PSMB8 mRNA, TPP2 mRNA. Moreover, we wanted to establish relationships between these molecules and clinical parameters.

## 2. Results

### 2.1. Concentrations of IVL, hBD-2 and Relative Expressions of IVL mRNA, hBD-2 mRNA, TPP2 mRNA, PSMB8 mRNA in Skin Biopsies from Patients with PV, AD and C

In all examined tissues, both proteins (IVL and hBD-2) and a relative expression of mRNA for IVL, hBD-2, TPP2, PSMB8 were detected (Appendix A).

The values of the tested parameters, except for mRNA for TPP2 and mRNA for PSMB8, were higher in the group of patients with inflammatory dermatoses than in healthy controls. In inflammatory lesions (all AD and PV biopsies), the median protein level was around 26 ng/mg for IVL and 1.5 ng/mg for hBD-2, whereas in the control group it was close to 0 for both proteins (Figure 1a,b). Concentrations of IVL, hBD-2 and the relative expression of hBD-2 mRNA and IVL mRNA were the highest in patients with PV (the average results demonstrated an approximately 1.6 and 4-fold increase in IVL mRNA and hBD-2 mRNA, respectively) and the lowest in C (near 0 for hBD-2 mRNA) (Figure 1a–d). The differences in IVL, hBD-2 at the protein level, hBD-2 mRNA, but not in IVL mRNA, were statistically significant among particular groups. TPP2 mRNA and PSMB8 mRNA values were the highest in C. TPP2 mRNA were the lowest in AD (Figure 1e). Differences between AD and C as well as PV and C in TPP2 mRNA values were statistically significant, but not between AD and PV. PSMB8 mRNA values were the lowest in PV (Figure 1f). Only differences between PV and C in PSMB8 mRNA values were statistically significant.

Key findings: The values of IVL and hBD-2 concentrations and hBD-2 mRNA were significantly statistically different between AD, PV and C. TPP2 mRNA values were statistically significantly different between AD and C and between PV and C, whereas values of PSMB8 mRNA were statistically significantly different only between PV and C.

### 2.2. Correlations between Tested Parameters

In all the biopsies, a strong positive correlation was found between concentrations of IVL and hBD-2 (*p* < 0.05) (Figure 2). The correlation was the strongest in the biopsies from patients with PV (Spearman’s R = 0.98, *p* = 1.65 × 10^−16^) (Figure 2c). This correlation was slightly weaker, but still very strong in biopsies from the patients with AD (Spearman’s R = 0.93, *p* = 1.27 × 10^−8^) (Figure 2b). The weakest correlation, although still strong, was in biopsies from C (Spearman’s R = 0.83, *p* = 0.003) (Figure 2a).

A statistically significant positive correlation was found between relative expression of TPP2 mRNA and PSMB8 mRNA in all the tissues (Figure 3).

TPP2 mRNA correlated positively with IVL mRNA in AD: Spearman’s R = 0.63, *p* = 0.006 and in PV: Spearman’s R = 0.60, *p* = 0.003 (Table 1). PSMB8 mRNA correlated positively with IVL mRNA only in AD: Spearman’s R = 0.55, *p* = 0.02 (Table 1).

No other correlation was found among IVL, hBD-2, IVL mRNA, hBD-2 mRNA, TPP2 mRNA and PSMB8 mRNA, neither in the patients with inflammatory skin diseases nor in the control group (Table 1).

Key findings: In all the biopsies, there were two positive correlations: -one between the hBD-2 and IVL concentrations,-and the other between PSMB8 mRNA and TPP2 mRNA.

### 2.3. Correlations of Tested Parameters and Clinical Data

In a cohort of patients, we analyzed correlations between expressions of studied proteins or mRNAs and some clinical data, such as age and gender of patients, duration of the disease, duration of a single lesion and itch intensity (Table 1). There were no correlations of the tested parameters with age and gender in either AD, or PV or C. No correlation was observed between the tested parameters (IVL, hBD-2, IVL mRNA, hBD-2 mRNA, TPP2 mRNA and PSMB8 mRNA) and a single lesion duration in AD or in PV, but only in PV patients did disease duration correlate positively with hBD-2 mRNA: Spearman’s R = 0.53, *p* = 0.01 (Table 1). Thus, in cases with the longest PV history, the hBD-2 mRNA expression was the strongest. On the contrary, there was an inverse correlation between IVL mRNA and skin itch intensity only in AD patients, assessing the patients with a 0–10 scale, where 10 meant the strongest sensation: *p* = 0.02; Spearman’s R = −0.53 (Figure 4). Thus, the most intense itch in AD was accompanied with the lowest IVL mRNA expression.

Key findings: In atopic dermatitis itch intensity correlated inversely with IVL mRNA.

### 2.4. Correlations of Tested Parameters with Skin Lesions Assessment

The correlations of all the tested parameters (IVL, hBD-2, IVL mRNA, hBD-2 mRNA, TPP2 mRNA and PMB8 mRNA), with an extension of skin lesions measured with the BSA scale, and disease severity assessed with EASI, SCORAD and modified SCORAD scales in AD, were evaluated. There were inverse correlations of hBD-2 mRNA with both SCORAD: *p* = 0.02; Spearman’s R = −0.53 and modified SCORAD scales: *p* = 0.01; Spearman’s R= −0.60 (Figure 5). Thus, the highest SCORAD or modified SCORAD scales’ results were in cases with the lowest hBD2 mRNA expression.

PSMB8 mRNA form AD biopsies correlated positively with the BSA scale results (Figure 6). Thus, in AD patients with more widely distributed lesions, there was a higher expression of PSMB8 mRNA. Interestingly, there was no relation of tested parameters with neither erythema (ESI-E), nor edema/papulation (ESI-I), nor excoriation (ESI-Ex), nor lichenification (ESI-L) values, nor their sum (ESI) within chosen lesions, nor with the results of EASI scale assessment (Table 1).

In the PV patients, hBD-2 mRNA positively and significantly correlated with erythema (E-PSI): *p* = 0.03; Spearman’s R = 0.45 and infiltration (I-PSI): *p* = 0.05; Spearman’s R = 0.43 and desquamation (S-PSI): *p* = 0.03; Spearman’s R = 0.46 and sum of erythema, induration and desquamation (PSI): *p* = 0.01; Spearman’s R = 0.56 within the selected biopsy lesions. Additionally, PSMB8 mRNA correlated positively with I-PSI: *p* = 0.03; Spearman’s R = 0.45. There was no correlation of the tested parameters with either BSA, or with PASI scales in PV (Table 1).

Key findings:-In AD, hBD-2 mRNA inversely correlated with SCORAD and modified SCORAD scales, and PSMB8 mRNA correlated positively with BSA.-In PV, hBD-2 mRNA correlated positively with erythema, infiltration and desquamation, and PSMB8 mRNA correlated positively with infiltration in the biopsied lesions.

### 2.5. Algorithm to Differentiate AD and PV

Based on the results of the tested parameters, a neurological network was built to differentiate biopsies form AD, PV and C tissues. Among the tested parameters, hBD-2 mRNA, PSMB8 mRNA and IVL and IVL mRNA were the most useful ones. The division between healthy subjects and patients with inflammatory disorders was carried out faultlessly based on the hBD-2 mRNA values (Figure 7). With only these values, it was possible to differentiate most cases of AD and PV, but there were two patients with PV allocated by the algorithm as AD. The exact division for AD, PV and the control group was possible only by extending the segregation criteria for the IVL and IVL mRNA values (Figure 7; left lower part of the diagram).

The differentiation between AD and PV was more exact when hBD-2 mRNA and PSMB8 mRNA expressions were considered for analysis (Figure 7; right lower part of the diagram).

According to the results of the algorithm built with the help of the neurological network, the relative expression of hBD-2 mRNA and of PSMB8 mRNA turned out to be the most valuable parameters in differentiating between the AD, PV and C biopsies. An ROC curve analysis of hBD-2mRNA and PASMB8 mRNA confirmed that hBD-2 mRNA is the best parameter to discriminate inflamed from healthy tissues, with the highest sensitivity and specificity. PSMB8 mRNA was more helpful than hBD-2 mRNA in differentiating between AD and PV biopsies, although with similar sensitivity and specificity (Figure 8).

Key findings: hBD-2 mRNA and PSMB8 mRNA are the most valuable parameters in differentiation of AD, PV and C biopsies.

## 3. Discussion

AD and PV usually can be properly recognized due to the characteristic clinical findings. Sometimes, the clinical presentation of those diseases may be confusing (Figure 9). Several recently shared data suggest that both AD and PV are complexes of distinct abnormalities rather than one disorder [5,47,48,49,50,51,52]. Additionally, there are rare cases of the coexistence of atopic and psoriatic changes in one patient [53,54,55,56], although usually one disease excludes the other due to the polarized interleukin milieu, cytokines produced mostly by Th2 cells in AD and Th 17, Th1 in PV [57,58]. Although, in long lasting AD lesions, Th1 and Th 17 cells are also involved [46]. The so-called “flip-flop” phenomenon, in which one disease turns into another, has been described by dermatologists for psoriasis and atopic dermatitis. This phenomenon is increasingly being observed, especially in connection with the use of biological drugs [59]. The IL-4 and IL-13 blocks stimulate PV in AD patients, while the IL-17 block promotes the development of AD in patients with PV [60,61]. Studies comparing both diseases improved our knowledge of inflammatory diseases’ pathophysiology leading to the development of new drugs, but still objective tools for diagnosis and differentiation are needed [62,63]. For the analysis, we took six parameters at the level of protein (IVL, hBD-2) and relative mRNA expression (IVL mRNA, hBD-2 mRNA, TPP2 mRNA, PSMB8 mRNA). In the first stage, we checked if the values of these six parameters differ between biopsies of inflammatory and healthy skin, and also between AD and PV lesions.

Atopic dermatitis lesions:A—an elbow fossa—classical atopic presentation with lichenification;B—an inner surface of the wrist—lesions are sharply demarcated like in psoriasis, but lichenification and scratch marks due to intense pruritus are typical for atopic changes;C—same scaling on atopic lesions, scales are thinner than in psoriasis;D—dermoscopic picture of AD lesion with irregularly distributed dilated vessels.

Psoriatic lesions:E—confluent psoriatic papules in an elbow fossa resembling atopic dermatitis, but no scratch marks and lichenification are visible;F—psoriatic lesions resembling atopic dermatitis with discrete scaling;G—typical psoriatic plaques covert with some scales, lesions are sharply demarcated;H—classical dermoscopic presentation of psoriasis with regular vessels and some scales.

Only patients with clinically evident and typically either psoriatic or atopic dermatitis lesions were included in our trial (Table 2). Increased levels of hBD-2 in biopsies from patients with PV compared to AD or the control group have been described, but there were some conflicting data concerning the comparison of those values between AD and the control group [64,65]. Those differences may be partly explained by the dissimilarity of the including criteria. Kim et al. collected biopsies from the patients who had not been treated with topical corticosteroids or calcineurin inhibitors for at least one week, while in our study patients had not been treated topically for one month [42]. Likewise, the biopsies in AD patients were taken from early lesions lasting up to two days, whereas in our study biopsies were mainly taken from long lasting lesions, mimicking that of psoriatic ones. In contrast to this study, Kim et al. reported that involucrin concentrations were decreased both in AD lesions and in healthy looking skin in comparison with the control group. One of the objectives of our study was to analyze tested parameters (IVL, hBD-2, IVL mRNA, hBD-2 mRNA, TPP2 mRNA and PMB8 mRNA) in relation to the evaluation of skin lesions. In our trial, the amount of hBD-2 and hBD-2 mRNA and IVL mRNA were significantly higher in biopsies from AD than from C. The relative expression of mRNA for hBD-2 seemed to be a useful clinical parameter in both AD and PV, as it corelated with the SCORAD and modified SCORAD severity scales in AD and with E-PSI, I-PSI, S-PSI, PSI in the evaluation of PV lesions. Similar correlations with SCORAD were observed by Clausen when hBD-2 was assessed in epidermis collected with the minimal invasive tape-strapping method [44]. The relative expression of mRNA for hBD-2 may be of value in clinical trials, especially with topical therapies, as it brings a helpful additional objective assessment of psoriatic plaques. So far, visual analog subjective scales have been used for describing AD or PV severity. Unfortunately, those descriptions may not be repeatable, because they can be influenced by several external factors such as temperature or light intensity.

Interestingly, we found positive correlations between IVL and hBD-2 concentrations in the skin of all the participants, which may suggest more constitutional type of hBD-2 expression (Figure 2). So far, the lower concentration of hBD-2 in patients with AD has been explained by the negative influence of Th2-derived cytokines, such as IL-4 and IL-13 [43]. It is known that the concentration of IVL in the skin of patients with PV or patients with AD is higher than in the skin of healthy controls. In the skin of patients with AD, there is a lower involucrin concentration than in the skin of patients with PV [66]. The results of our study confirm that tendency. Differences in involucrin concentrations, but not in IVL mRNA, were statistically significant between AD and PV.

An itching sensation was present in both studied diseases. Among our patients with inflammatory skin diseases, all except one patient with PV reported pruritus with different intensity (Appendix A). There are clinical observations that many, if not most PV patients, complain about the pruritus of skin lesions. For this reason, pruritus cannot be used to differentiate between these two diseases [67]. Pruritus is one of the diagnostic criteria in AD and it is clinically stronger in AD than in PV, where it can be absent. Properly developed stratum corneum acts as a shield from the environment factors covering itch receptors. Inflammation and abnormal keratinization may lead to nerve ending irritation. As involucrin is involved in the keratinization process, we checked whether it was related with itch sensation in our patients and we found that IVL mRNA, as the only parameter, correlated with itch intensity in AD. The correlation was negative, indicating a stronger itch sensation when IVL mRNA expression was lower (Figure 4).

Further experiments are needed to determine whether IVL mRNA could be useful for validation of pruritus in patients with atopic dermatitis.

The inflammation marker—hBD-2 mRNA (human β-defensin-2)—was elevated in patients with the longest PV history (where inflammation was the highest), while cases with highest clinical scores (SCORAD) were marked with decreased hBD-2 mRNA expression, suggesting impaired inflammatory response in AD patients.

The involvement of involucrin or hBD-2 in the pathogenesis of AD and PV is obvious. A novelty in our clinical approach is the consideration of the relative expression of TPP2 mRNA and mRNA for PSMB8 (the proteolytic inducible iB5 subunit of 20S proteasome). Tripeptidyl peptidase 2 is involved in the regulation of HIV responses [46], but the role of TPP2 in inflammatory skin diseases has not yet been understood. Recently, a family was described with deletion of *TPP2* gene, where either cognitive impairment or immune system imbalance was dominant, with a tendency for the development of inflammatory skin diseases, including AD. Here, we aimed to check out if the expression of TPP2 at mRNA level is altered in AD or PV skin. In both diseases, TPP2 mRNA is decreased in comparison to the healthy skin. A similar tendency is observed with inducible subunit of the immunoproteasome. Since both 20S proteasome and TPP2 are active in the same protein degradation pathway, this result is not surprising [68,69]. In AD and PV epidermal cells, turnover is increased. Especially in PV, immature cells reach the skin surface. As the role of keratinocytes and corneocytes is to maintain a proper skin barrier, they produce more proteins such as involucrin or defensin to prevent barrier disruption, but they are probably not mature enough or they do not have sufficient time for other, more complicated proteins, such as TPP2 and PSB8. It seems that these cells are primarily programed to protect the barrier. For TPP2, most are probably post-transcriptional modifications, in particular the assembly and disassembly of this giant protease, with a twist—it may contain anywhere from two stacked rings to numerous stacked rings—which changes its proteolytic productivity. Our results need further studies to corroborate them, also involving changes in TPP2 at the post-translational level.

Immunoproteasome is an important molecule that participates in the presentation of autoantigens to the immune system. The regulatory molecule of the PA28 immunoproteasome, composed of the alpha and beta subunits, accelerates the transport of autoantigens and increases the pool of epitopes for presentation to the immune system. The presence of the inducible proteolytic iB1, iB2 and iB5 subunits fosters the generation of the antigens presented by MHC I to the immune system. The iB5 (PSMB8) sub-unit is particularly important, because it is responsible for the chymotrypsin-like activity of the proteasome. An increased expression of the immunoproteasome is associated with rheumatic diseases [70], and arises in response to certain viral and bacterial infections; however, its association with the clinics of inflammatory skin diseases has not yet been studied. Bortezomib, carfilzomib, orpozomib, ixazomib KZR-616 inhibitors of proteasome and immunoproteasome [45] are in clinical use as anticancer drugs. The epoxyketone inhibitor KZR-616, which selectively targets the immunoproteasome, is currently being evaluated in phase 2 clinical trials in patients with autoimmune disorders, including lupus nephritis (LN), dermatomyositis (DM), and polymyositis (PM) [45]. We aimed to investigate if alternations in immunoproteasome expression are found in specimens from AD or PV patients. The most interesting and statistically significant relations for TPP2 mRNA and PSMB8 mRNA are presented on Figure 10. There was a statistically significant positive correlation between TPP2 mRNA and PSMB8 mRNA in all groups (C, AD and PV). Expressions of TPP2 mRNA and PSMB8 mRNA were the highest in the control group. TPP2 mRNA expression was the lowest in AD, while PSMB8 mRNA expression was the lowest in PV. It is well known that TPP2 is assembled and disassembled depending on metabolic status—thus, further studies are needed (including TPP2 visualization by atomic force electron microscopy—for examining forms of TPP2 in particular skin biopsies). Also, PSMB8 might be post-translationally regulated in the cytoplasm. Interestingly, PSMB8 mRNA correlated positively with the area of the skin lesions (BSA) in AD and with the thickness of lesions due to inflammation in PV (I-PSI). These results suggest the possible clinical usefulness of PSMB8 in inflammatory skin diseases in the diagnostic, or maybe also therapeutic, process.

In the second step of our work, with the help of an artificial network, we tried to find a diagnostic algorithm. The results of hBD-2 mRNA, IVL, IVL mRNA and PSMB8 mRNA turned out to be the most valuable. hBD-2 mRNA results alone were sufficient to exclude healthy skin biopsies from inflammatory lesions, but not to divide AD from PV faultlessly. Although hBD-2 mRNA results helped to distinguish most AD and PV biopsies, there were seven biopsies in which additional parameters were necessary for further diagnostics. It turned out that there are two possibilities: one with values of IVL and IVL mRNA and the other, more simple, with PSMB8 mRNA results. We have not found such a conclusion in the literature so far. Thus, in controversial cases, the level of PSMB8 mRNA expression might help in distinguishing difficult cases and introducing exact treatment.

A novelty of our work is the simultaneous study of the constitutive subunit of the proteasome and TPP2 (both are intracellular markers associated with antigen turnover in the cytoplasm) together with the analysis of involucrin (cellular differentiation in the skin) and human beta defensin-2 (a marker of inflammation mainly in psoriasis) and their correlation with selected clinical parameters.

As far as we know, there are no such clinical tests based on a similar analysis. Its application in the clinic would be quite complicated. More research is needed to, for example, try to differentiate non-obvious cases of AD and PV in this way.

The limitation of our work is the small amount of carefully selected cases with clear diagnosis of either psoriasis or atopic dermatitis. The results obtained indicate the possibility of using the chosen parameters in the process of differentiating AD and PV, if only one had an approved diagnostic test for clinical practice and not, as in the case of our study, an experimental method based on the measurement of relative mRNA expression. Our work shows some tendencies and important relations, and can inspire other scientists both in diagnostic and in therapeutic fields.

## 4. Materials and Methods

### 4.1. Participants

There were 51 adult participants: 41 patients diagnosed with either atopic dermatitis or psoriasis vulgaris and 10 volunteers, who represented the control group. 19 subjects with AD (including 11 women and 8 men) and 22 patients with PV (9 women and 13 men) were included in the study. They were recruited from the patients hospitalized in the years 2013–2015 in the Department of Dermatology of Military Institute of Medicine in Warsaw. Use of any systemic or topical treatments within the last month or any uncontrolled systemic disease, including systemic neoplastic processes, were excluded. The majority of the participants had not been treated for inflammatory diseases with systemic therapies until the recruitment. Out of 51 participants, only two psoriatic patients had received systemic therapies before the recruitment. The first one, a 30-year-old man, had taken cyclosporine for three months four years earlier. The second one, a 25-year-old woman, had been treated with methotrexate for four months and then acitretin for five months, which she had finished six months before the recruitment. Control biopsies were obtained from healthy skin of volunteers during surgical operation of stomach reduction. Only patients with negative personal and family history for both AD and PV were qualified for the control group. The characteristics of patient cohort are displayed in Table 2. Before signing the informed consent form, all study procedures and concerns were sufficiently discussed with each participant. The study was approved by the Ethics Committee of Military Institute of Medicine in Warsaw and was conducted according to the Declaration of Helsinki.

### 4.2. Clinical Assessment of Skin Lesions

In all patients, the extent of the lesions was assessed by the BSA scale, and medium itch severity from last 3 days was evaluated using the numeric scale from 0 to 10, where 10 indicated the most aggravated itch.

The severity of AD was measured with the EASI, SCORAD and modified SCORAD (without a subjective assessment of itch and sleep disturbances) scales, while the severity of PV was described by the PASI scale.

The site of the biopsy was assessed for the intensity of erythema ESI-E (0–3), edema/papulation ESI-I (0–3), excoriation ESI-Ex (0–3) and lichenification ESI-L (0–3) and their sum ESI (0–12) in all patients with AD, and the intensity of erythema E-PSI (0–3), infiltration I-PSI (0–3) and desquamation S-PSI (0–3) and their sum PSI (0–9) in all patients with PV.

### 4.3. Skin Biopsies

A representative skin lesion for AD or PV was anesthetized with ethyl chloride for a few seconds and a 5 mm punch biopsy was taken and divided into two parts. One part was frozen immediately and the other was frozen after placing it in the 350 μL lysis buffer from NucleoSpin^®^RNA II kit (Macherey-Nagel, Duren, Germany). The tissues were stored at −80 °C until the proper number of biopsies was collected.

### 4.4. Tissue Homogenate Preparation and ELISA

Tissue samples were rinsed in ice-cold phosphate-buffered saline (PBS), weighed, immediately homogenized in PBS (50 mg tissue in 1 mL of buffer) using a motor driven homogenizer (BT Lab Systems, Saint Louis, MO, USA) and centrifuged (3000 rpm for 15 min). The supernatant of the homogenate was aliquoted and stored at −80 °C. Levels of involucrin (CSB-E09332H) and beta-defensin-2 (CSB-E13201H) in homogenates of skin were determined using enzyme-linked immunosorbent assay (ELISA) kits according to the manufacturer’s guidelines (both kits from Wuhan Huamei Biotech Co., Ltd., Wuhan, China).

### 4.5. Total RNA Isolation, Reverse Transcription (RT) and Real Time PCR

RNA from tissue samples was isolated with NucleoSpin^®^RNA II kit (Macherey-Nagel, Duren, Germany), according to the manufacturer’s protocol. The quality and concentration of RNA was assessed with NanoDrop spectrophotometer. The RT was performed with High Capacity RNA-to-cDNA Kit according to the manufacturer’s protocol (Applied Biosystems, ThermoFisher Scientific, Waltham, MA, USA). cDNA was stored at −20 °C. Gene expression was measured with the relative quantitation (RQ) using a comparative cycle threshold (CT). Real Time PCR was performed in Abi Prism 7500 (Applied Biosystems, ThermoFisher Scientific, Waltham, MA, USA) in 96-well optical plates. Each sample was run in triplicate and supplied with endogenous control (human GAPDH no. Hs01032443_m1) as previously described by Kim et al. [42]. For gene expression, TaqMan Expression Assays were used: TPP2: Hs01557058_m1; PSMB8: Hs00544760_g1; DEFB4A: Hs00175474_m1 and IVL: Hs00902520_m1. All probes were stained with FAM (all from Applied Biosystems, ThermoFisher Scientific, Waltham, MA, USA). Reactions were run in 25 μL volume with TaqMan Universal Master Mix (Applied Biosystems, ThermoFisher Scientific, Waltham, MA, USA), an appropriate primer set, an MGB (minor groove binder) probe and 5 ng of cDNA template. Universal thermal conditions, that is, 10 min at 95 °C, and 40 cycles of 15 s at 95 °C and 1 min at 60 °C, were used. Data analysis was carried out with sequence detection software version 1.2 (Applied Biosystems, ThermoFisher Scientific, Waltham, MA, USA).

### 4.6. Statistical Analysis

Statistical analysis was carried out in StatSoft, Inc. (2014), STATISTICA (data analysis software system), version 12. Tulsa, OK, USA

The data were analyzed by using the Mann–Whitney U test. Correlations were determined by using Spearman’s rank correlation coefficient (Spearman’s R). Significance was assumed at a *p* value of 0.05 or less.

### 4.7. Classification Algorithm Based on the Test Results

Classification trees and artificial network were applied in order to check whether the obtained data can be used for classifying the skin biopsies into the groups AD, PV and C. An artificial network was built with the STATISTICA application. The neural network was taught to classify biopsies into AD, PV and C upon receiving the results of randomly selected cases, approximately 30 percent of them. The operation was verified with the results of 30 percent of all the cases, randomly selected, excluding the cases used for the teaching. Finally, the network was checked by classifying the remaining cases. Based on the results obtained, a decision algorithm was built—for qualification of patients to AD or PV.

## 5. Conclusions

In the examined tissues, values of hBD-2 mRNA, IVL mRNA and PSMB8 mRNA were the most interesting parameters. hBD-2 mRNA was solely sufficient to distinguish inflamed lesions from healthy skin, and in most cases to separate AD and PV biopsies. Additionally, hBD-2 mRNA correlated positively with SCORAD and modified SCORAD in AD and with the assessment of a single skin lesion (PSI, E-PIS, I-PSI, L-PSI) in PV.

IVL mRNA, in AD changes, correlated negatively with itch sensation. PSMB8 mRNA also seems to be of clinical value, as it positively correlated with the extent of lesions in AD and with inflammatory infiltrate within changes in PV (I-PSI). Further examinations are needed to determine if these findings are of clinical value. 

It would also be interesting to check if a PSMB8 blockade could diminish inflammation in psoriatic and atopic dermatitis lesions.

## Figures and Tables

**Figure 1 ijms-25-09192-f001:**
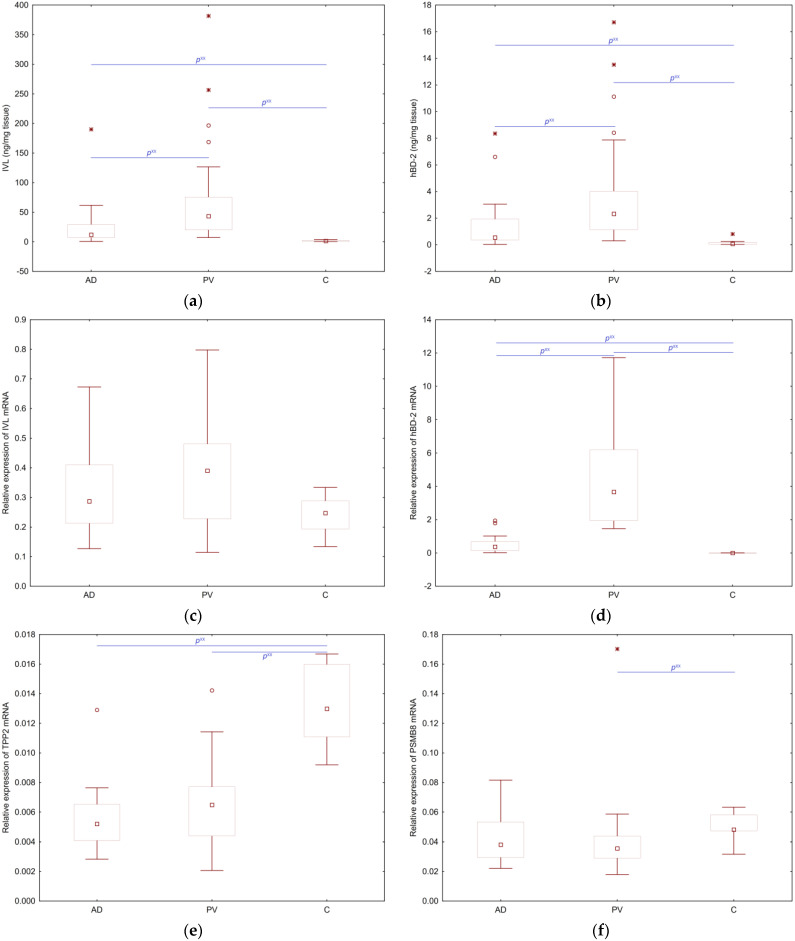
Results of tested parameters in: AD—patients with atopic dermatitis, PV—patients with psoriasis vulgaris, C—healthy volunteers; (**a**) involucrin concentrations: differences statistically significant between AD and C (*p* = 0.00011), PV and C (*p* = 0.000009) and PV and AD (*p* = 0.002); (**b**) human β-defensin-2 concentrations: differences are statistically significant between AD and C (*p* = 0.00122), PV and C (*p* = 0.000022) and PV and AD (*p* = 0.008); (**c**) relative expression of mRNA for involucrin: differences are not statistically significant between AD and C (*p* = 0.148369), PV and C (*p* = 0.058670) and PV and AD (*p* = 0.395); (**d**) relative expression of mRNA for human β-defensin-2: differences are statistically significant between AD and C (*p* = 0.00002), PV and C (*p* = 0.000009) and PV and AD (*p* < 0.001); (**e**) relative expression of mRNA for TPP2: differences between AD and C (*p* = 0.00004) and PV and C (*p* = 0.00010) values are statistically significant, but not between AD and PV values. (**f**) relative expression of mRNA for PSMB8: only differences between PV and C are statistically significant (*p* = 0.0047). *p*^xx^ means *p* < 0.01, 

 stands for the extreme value, 

 denotes an outlier.

**Figure 2 ijms-25-09192-f002:**
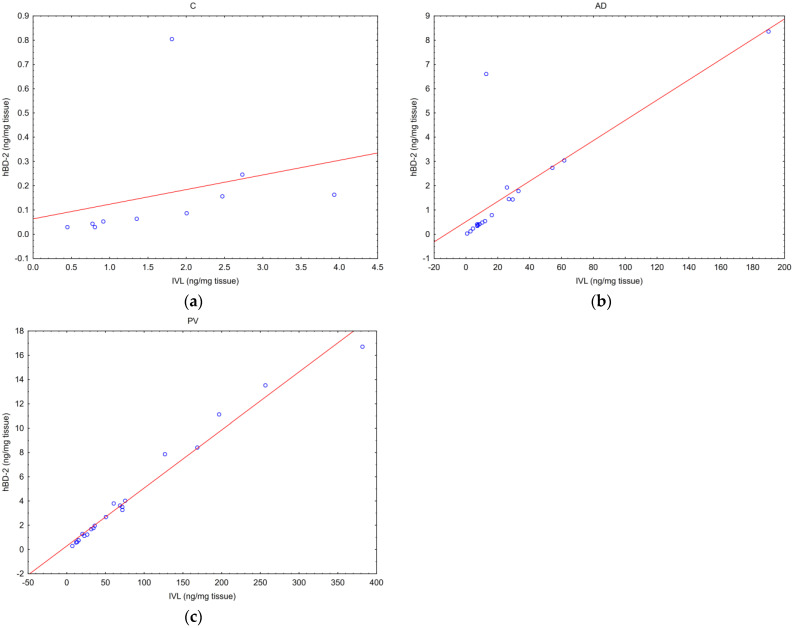
Correlation of concentrations of human β-defensin-2 and involucrin in: (**a**) healthy skin (*p* = 0.003, Spearman’s R = 0.89); (**b**) atopic dermatitis lesions (*p* < 0.001, Spearman’s R = 0.93); (**c**) psoriatic lesions (*p* < 0.001, Spearman’s R = 0.98).

**Figure 3 ijms-25-09192-f003:**
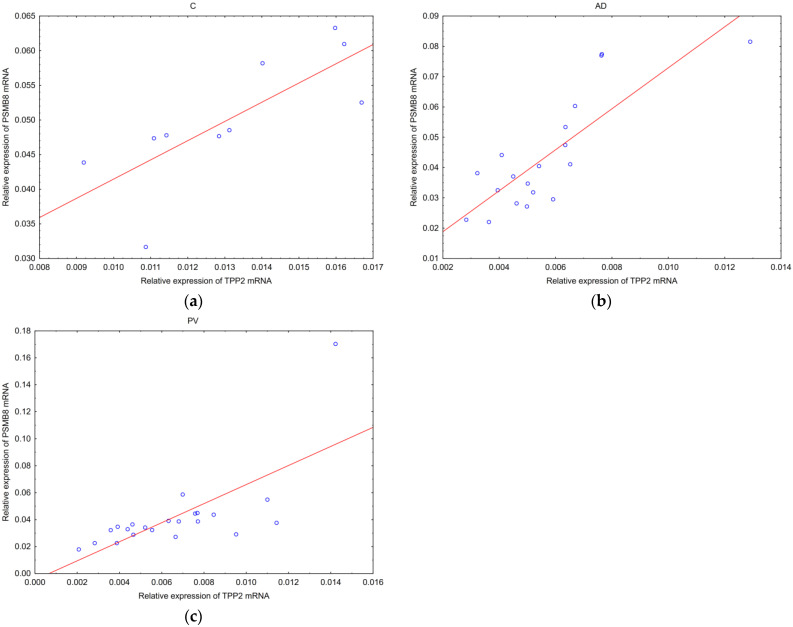
Correlation of relative expression of PSMB8 mRNA and relative expression of TPP2 mRNA in: (**a**) healthy skin (*p* = 0.0005, Spearman’s R = 0.89); (**b**) atopic dermatitis lesions (*p* = 0.0001, Spearman’s R = 0.77); (**c**) psoriatic lesions (*p* = 0.0002, Spearman’s R = 0.72).

**Figure 4 ijms-25-09192-f004:**
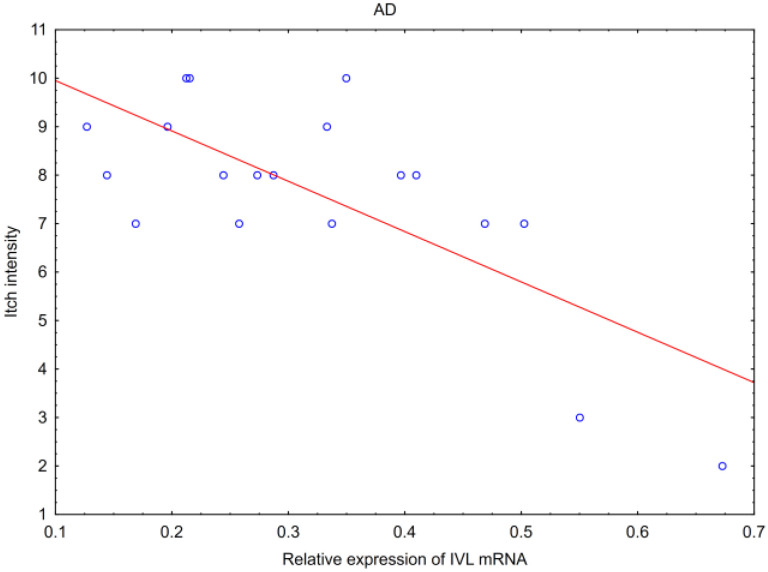
Correlation of IVL mRNA with intensity of itch sensation in patients with atopic dermatitis (*p* = 0.02, Spearman’s R = −0.53).

**Figure 5 ijms-25-09192-f005:**
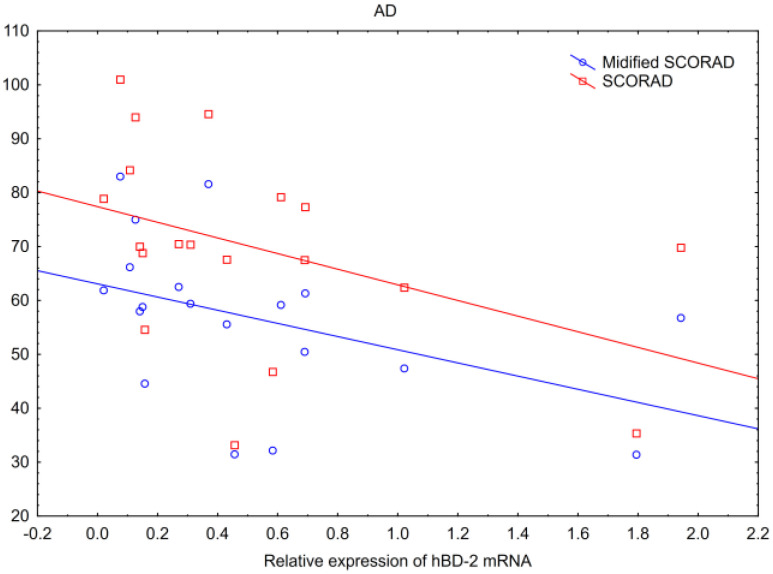
Correlation of hBD-2 mRNA with SCORAD (red) (*p* = 0.02, Spearman’s R= −0.53) and modified SCORAD (blue) (*p* = 0.01, Spearman’s R= −0.60) in patients with atopic dermatitis.

**Figure 6 ijms-25-09192-f006:**
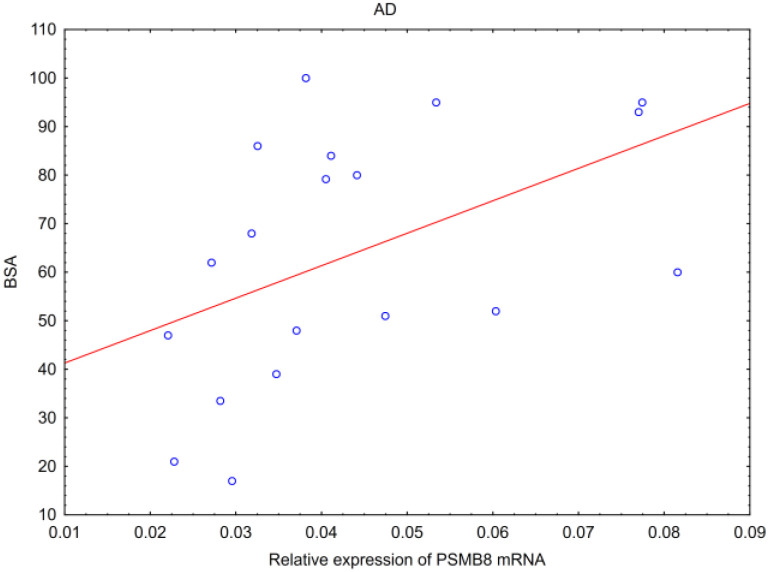
Correlation of PSMB8 mRNA with BSA (*p* = 0.0128, Spearman’s R = 0.559) in patients with atopic dermatitis.

**Figure 7 ijms-25-09192-f007:**
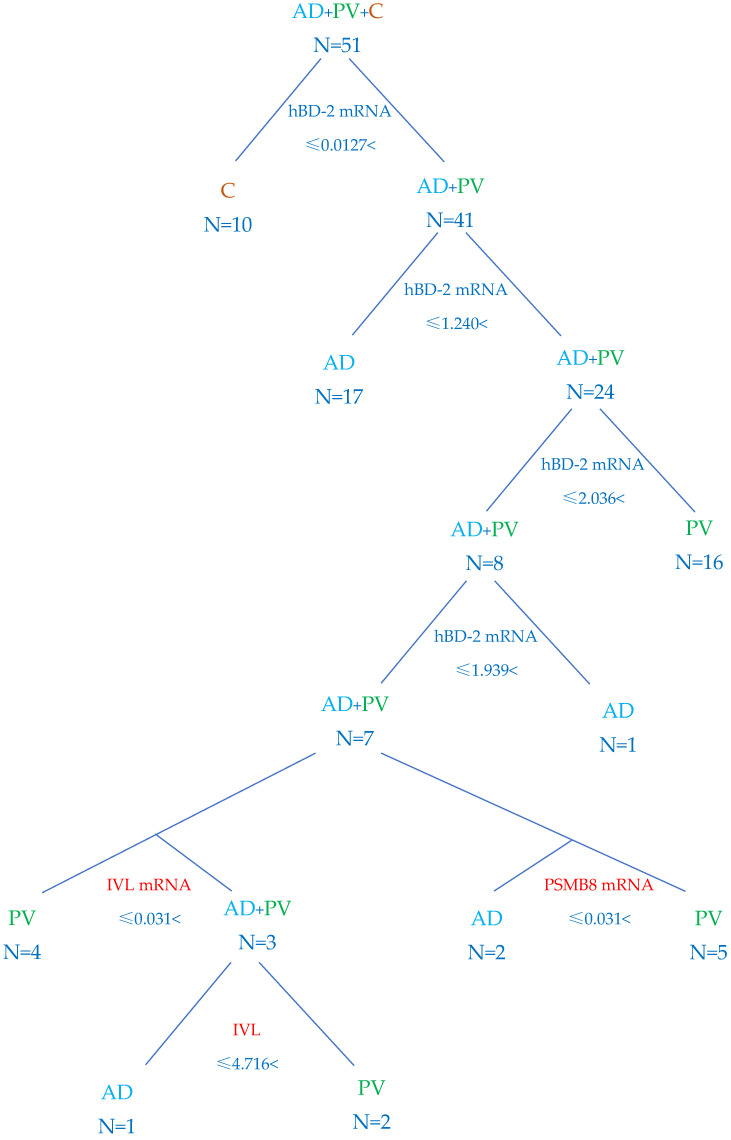
A diagnostic algorithm based on the values of hBD-2 mRNA, IVL, IVL mRNA and PSMB8 mRNA. Classification of biopsies into groups of healthy skin and inflammatory skin diseases was completed based on the results of hBD-2 mRNA (first division). Classification of biopsies into groups of AD, PV and C based on the results of hBD-2 mRNA, IVL and IVL mRNA (**left lower part of the diagram**) or based on the results of hBD-2 mRNA and PSMB8 mRNA (**right lower part of the diagram**).

**Figure 8 ijms-25-09192-f008:**
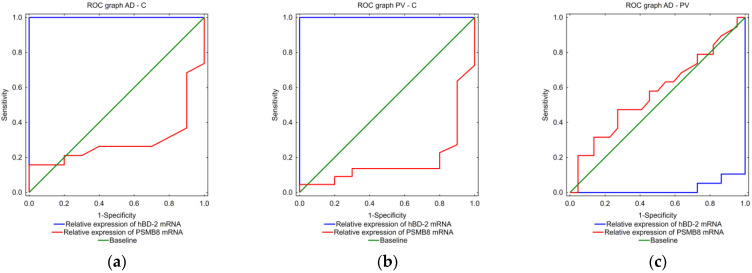
The ROC graphs for hBD-2 mRNA and PSMB8 mRNA for results in: (**a**) AD and C biopsies, (**b**) PV and C biopsies, (**c**) for AD and PV biopsies.

**Figure 9 ijms-25-09192-f009:**
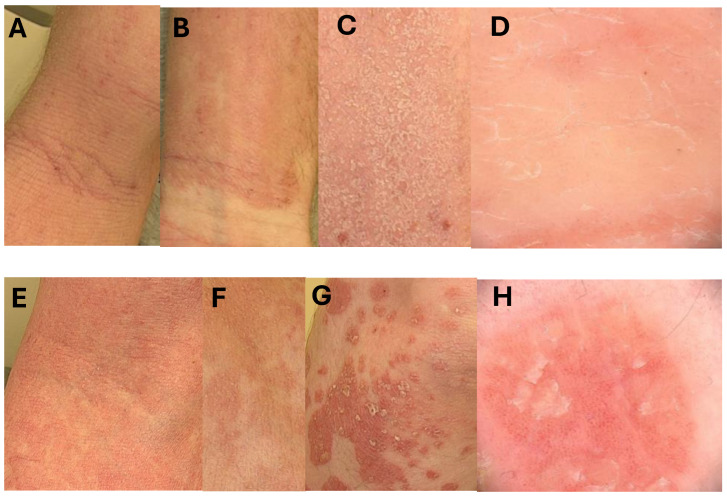
Clinical manifestations of atopic dermatitis (**A**–**D**) and psoriasis vulgaris (**E**–**H**)—lesions that may cause confusion in diagnosis.

**Figure 10 ijms-25-09192-f010:**
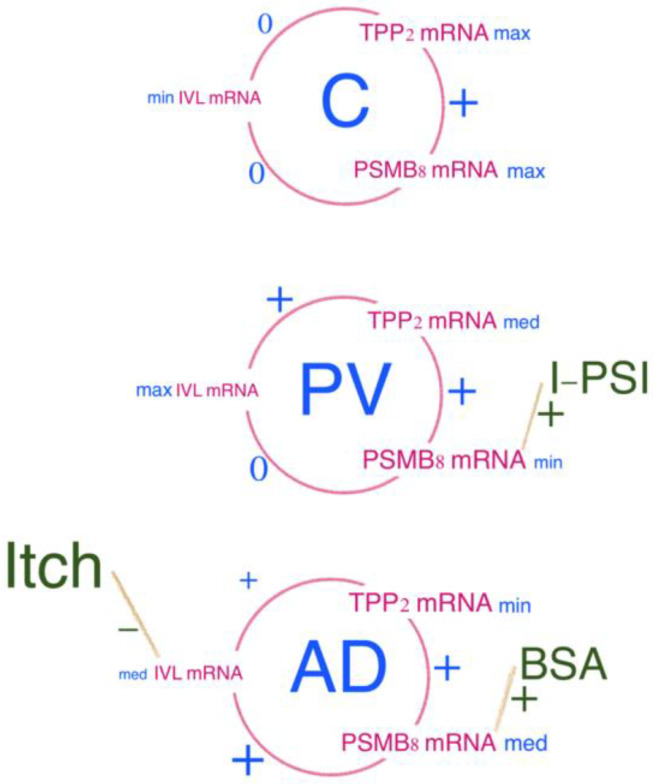
Schematic presentation of correlations between TPP2 mRNA, PSMB8 mRNA and IVL mRNA and significantly related clinical features in groups of AD, PV and C. In all the patients, TPP2 mRNA corelated positively with PSMB8 mRNA, suggesting the cooperation of both molecules in the skin, in any stage. PSMB8 mRNA correlated positively with IVL mRNA only in an inflammation state, and was elevated in both AD and PV, but not in healthy skin. PSMB8 mRNA corelated positively with skin lesions extension (BSA) in AD and with inflammation in a single lesion (I-PSI) in PV. min—minimal values, med—medial values, max—max values.

**Table 1 ijms-25-09192-t001:** Statistical analysis of IVL and hBD-2 concentrations and relative expression of IVL mRNA, hBD-2 mRNA, TPP2 mRNA, PSMB8 mRNA and selected clinical parameters in biopsies from AD, PV and C.

	IVL	hBD-2	IVL mRNA	hBD-2 mRNA	TPP2 mRNA	PSMB8
IVL	AD NA	AD *p* ** R = 0.93	AD NS	AD NS	AD NS	AD NS
	PV NA	PV *p* ** R = 0.98	PV NS	PV NS	PV NS	PV NS
	C NA	C *p* ** R = 0.83	C NS	C NS	C NS	C NS
hBD-2	AD *p* ** R = 0.93	AD NA	AD NS	AD NS	AD NS	AD NS
	PV *p* ** R = 0.98	PV NA	PV NS	PV NS	PV NS	PV NS
	C *p* ** R = 0.83	C NA	C NS	C NS	C NS	C NS
IVL mRNA	AD NS	AD NS	AD NA	AD NS	AD *p* ** R = 0.67	AD *p* * R = 0.56
	PV NS	PV NS	PV NA	PV NS	PV *p* ** R = 0.60	PV NS
	C NS	C NS	C NA	C NS	C NS	C NS
hBD-2 mRNA	AD NS	AD NS	AD NS	AD NA	AD NS	AD NS
	PV NS	PV NS	PV NS	PV NA	PV NS	PV NS
	C NS	C NS	C NS	C NA	C NS	C NS
TPP2 mRNA	AD NS	AD NS	AD *p* ** R = 0.67	AD NS	AD NA	AD *p* ** R = 0.77
	PV NS	PV NS	PV *p* ** R = 0.60	PV NS	PV NA	PV *p* ** R = 0.72
	C NS	C NS	C NS	C NS	C NA	C *p* ** R = 0.89
PSMB8 mRNA	AD NS	AD NS	AD *p* ** R = 0.67	AD NS	AD *p* ** R = 0.77	AD NA
	PV NS	PV NS	PV NS	PV NS	PV *p* ** R = 0.72	PV NA
	C NS	C NS	C NS	C NS	C *p* ** R = 0.89	C NA
Age Gender	AD NS					
	PV NS					
	C NS					
Disease duration	AD NS	AD NS	AD NS	AD NS	AD NS	AD NS
	PV NS	PV NS	PV NS	PV *p* * R = 0.53	PV NS	PV NS
Single lesion duration	AD NS					
	PV NS					
Itch	AD NS	AD NS	AD *p* * R = −0.53	AD NS	AD NS	AD NS
	PV NS	PV NS	PV NS	PV NS	PV NS	PV NS
BSA	AD NS	AD NS	AD NS	AD NS	AD NS	AD *p* * R = 0.56
	PV NS	PV NS	PV NS	PV NS	PV NS	PV NS
SCORAD	AD NS	AD NS	AD NS	AD *p* * R = −0.53	AD NS	AD NS
Modified SCORAD				AD *p* * R = −0.60		
EASI	AD NS	AD NS	AD NS	AD NS	AD NS	AD NS
ESI						
ESI-E						
ESI-I						
ESI-L						
PSI	PV NS	PV NS	PV NS	PV *p* ** R = 0.56	PV NS	PV NS
E-PSI				PV *p* * R = 0.45		PV *p* *
I-PSI				PV *p* * R = 0.42		R = 0.45
S-PSI				PV *p* * R = 0.46		PV NS

AD—atopic dermatitis, PV—psoriasis vulgaris. C—controls, NA—not applicable, NS—not significant correlation, *p* * means *p* ≤ 0.05, *p* ** means *p* < 0.01, R– Spearman’s R

**Table 2 ijms-25-09192-t002:** Characteristics of patient cohort involved in this study.

**AD ^1^**	**Variable**	**Median**	**Minimum**	**Maximum**	**Bottom Quartile**	**Upper Quartile**
	Age	32	20	55	23	49
	Disease duration (days)	7665	62	15,330	3650	10,950
	Lesion duration (days)	62	2	1825	21	279
	Pruritus	8	2	10	7	9
	BSA	62%	17%	100%	47%	86%
	SCORAD	70.00	33.20	101.00	62.40	79.20
	Modified SCORAD	58.80	31.40	83.00	47.40	62.50
	EASI	35.80	10.10	62.70	22.50	50.50
	ESI	9	5	12	8	11
	ESI-E	3	1	3	2	3
	ESI-I	2	1	3	2	3
	ESI-Ex	2	1	3	1	3
	ESI-L	2	1	3	2	3
**PV ^1^**	**Variable**	**Median**	**Minimum**	**Maximum**	**Bottom Quartile**	**Upper Quartile**
	Age	36	25	58	28	44
	Disease duration (days)	5840	42	13,870	1825	10,220
	Lesion duration (days)	230.5	14.0	3650.0	124.0	730.0
	Pruritus	4	1	8	3	7
	BSA	19.55%	3.70%	76.50%	9.25%	29.00%
	PASI	17.15	2.60	43.90	9.30	20.20
	PSI	7	3	9	6	8
	E-PSI	3	1	3	2	3
	S-PSI	2	1	3	2	3
	I-PSI	2	1	3	2	2

^1^ AD—atopic dermatitis, PV—psoriasis vulgaris.

## Data Availability

Data are available on request due to restrictions, e.g., privacy or ethical considerations. The data presented in this study are available on request from the corresponding authors.

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
