# Peer review of "Human Beta Defensin-2 mRNA and Proteasome Subunit β Type 8 mRNA Analysis, Useful in Differentiating Skin Biopsies from Atopic Dermatitis and Psoriasis Vulgaris Patients"

_ijms, 2024, doi:10.3390/ijms25179192_

Round 1

Reviewer 1 Report

Comments and Suggestions for Authors

Dear authors 

To improve your manuscript I make the following suggestion:

-Introduction: Although the introduction cites several studies, it would be advantageous to include a more thorough analysis of the body of research regarding the similarities and differences between AD and PV. It would be stronger if the study's hypothesis or main research question were presented initially.

-Results: The detailed presentation, while thorough, can be overwhelming. Summarizing key findings more succinctly at the end of the section could enhance clarity.

-Discussion. A clearer focus on the results' novelty and possible significance would enhance the conversation. How do these findings expand the discipline or modify the clinical practice? A more detailed discussion of the limitations of the study design, sample size, and potential confounding factors would strengthen the validity of the conclusions.

Comments on the Quality of English Language

The manuscript is well-written with few grammatical errors. Focus on simplifying complex sentences, ensuring consistent verb tense, and refining the use of articles and prepositions. These minor adjustments will enhance the readability and professionalism of the text.

Author Response

Comment 1- Introduction: Although the introduction cites several studies, it would be advantageous to include a more thorough analysis of the body of research regarding the similarities and differences between AD and PV. It would be stronger if the study's hypothesis or main research question were presented initially.

We have modified introduction text:

Clinically AD may mimic PV, moreover primary diagnosed AD may transform into PV and vice versa There is an observation of coexistence of PV and AD features in the same inflammatory lesions, especially in children. These lesions are called PD (psoriasis dermatitis) or PsEma and with time can evolve into either AD or PV. [17].

Additionally histopathological examination may show both spongiosis and parakeratisis with hypogranulosis that make differential diagnosis impossible [18]. There is no commonly available and clinically approved test for differentiation between biopsies from AD and PV lesions.

According to literature [34-41] involucrin and human β-defensin-2 seemed to be good candidates for differentiation between atopic dermatitis and psoriasis vulgaris as their higher concentrations were described in psoriasis vulgaris. Neither TPP2 nor PSMB8 are recognized biomarkers in differentiating the diagnosis of AD and PV. However, since their participation in connection with these diseases has been described in few works [28,32], we decided to investigate whether these two molecules differ between AD, PV and controls. We decided to rely on the mRNA expression analysis of all proteins to be able to compare their expression at the mRNA level. Involucrin is a major marker of skin cell differentiation [36,39], and human beta defensin -2 is a marker of inflammation [37,40,41]. In turn, for the immunoproteasome, the activity - essential subunit is iB5 (PSMB8) [21,27,42] and for TPP2 - the most conserved protease - we chose the most frequently studied sequence to have a reference for other works [29,43]. Since the size of the sample that can be taken from the patient is limited - it was more reasonable to study the reference molecules at the mRNA level. Additionally, for markers of inflammation and skin differentiation - we studied expression also at the protein level to have a second point of reference.

The aim of our work was to verify if it is possible to differentiate tissues from AD, PV and C participants based on analysis of the six parameters: IV, hBD-2, IVL mRNA, hBD-2 mRNA, PSMB8 mRNA, TPP2 mRNA. Moreover we wanted to establish relationship between those molecules and clinical parameters.

Comment 2 -Results: The detailed presentation, while thorough, can be overwhelming. Summarizing key findings more succinctly at the end of the section could enhance clarity.

After each section we have written key findings:

  1. Values of IVL and hBD-2 concentrations and hBD-2 mRNA, were significantly statistically different between AD, PV and C. TPP2 mRNA values were statistically significantly different between AD and C and between PV and C, whereas values of PSMB8 mRNA were statistically significantly different only between PV and C.
  2. In all the biopsies there were two positive correlations: one between hBD-2 and IVL concentrations and the other between PSMB8 mRNA and TPP2 mRNA.
  3. In AD itch intensity correlated inversely with IVL mRNA.
  4. In AD hBD-2 mRNA inversely correlated with SCORAD and modified SCORAD scales and PSMB8 mRNA correlated positively with BSA
  5. In PV hBD-2 mRNA correlated positively with erythema, infiltration and desquamation and PSMB8 mRNA correlated positively with infiltration
  6. hBD-2 mRNA and PSMB8 mRNA are the most valuable parameters in differentiation of AD, PV and C biopsies.

Comment 3-Discussion. A clearer focus on the results' novelty and possible significance would enhance the conversation. How do these findings expand the discipline or modify the clinical practice? A more detailed discussion of the limitations of the study design, sample size, and potential confounding factors would strengthen the validity of the conclusions.

In discussion we have included the following text:

A novelty of our work is the simultaneous study of the constitutive subunit of the proteasome and TPP2 (both are intracellular markers associated with antigen turnover in the cytoplasm) together with the analysis of involucrin (cellular differentiation in the skin) and human beta defensin-2(a marker of inflammation mainly in psoriasis) and their correlation with selected clinical parameters.

As far as we know, there are no such clinical tests based on similar analysis. Its application in the clinic would be quite complicated. More research is needed to, for example, try to differentiate non-obvious cases of AD and PV in this way.

The limitation of our work is small amount of carefully selected cases with clear diagnosis of either psoriasis or atopic dermatitis The results obtained indicate the possibility of using the chosen parameters in the process of differentiating AD and PV, if only one had an approved diagnostic test for clinical practice and not, as in the case of our study, an experimental method based on the measurement of relative mRNA expression. Our work shows some tendencies and important relations and can inspire other scientists both in diagnostic and in therapeutic fields.

Reviewer 2 Report

Comments and Suggestions for Authors

In this manuscript, Terlikowska-Brzósko and colleagues examined the biomarker potential of Human beta defensin-2, IVL, PSM8, and TPP2 in distinguishing healthy skin from AD or PV. The authors used ELISA, qRT-PCR, and co-relation analysis to achieve this. The high point of the presented study is the correlation analysis with clinical factors. The study is nicely written and presented. The following areas of concern need to be addressed.

1) The genes selected for this study are significant in skin homeostasis and diseases. However, the criteria for presenting them as biomarkers need to be more detailed and clearly outlined.

2) Table 1 and Figure 1 are redundant and almost present the same information. 

3) Use ROC curve analysis to present the specificity and sensitivity of these genes in distinguishing healthy skin from AD or PV. 

4) Compare available clinical tests for AD and PV with these genes side by side to show the presented parameters are better or equivalent to currently used methods in the clinic. 

5) Itch is a demarcating feature of AD and is often not present in PV. Only a small fraction of PV patients present with itch. How many PV patients show itch as a clinical feature? Could itch be a distinguishing factor between healthy skin, AD, and PV?

6) Articles cited to show psoriasis prevalence are old. Use recent articles and data to show psoriasis prevalence. 

Author Response

In this manuscript, Terlikowska-Brzósko and colleagues examined the biomarker potential of Human beta defensin-2, IVL, PSM8, and TPP2 in distinguishing healthy skin from AD or PV. The authors used ELISA, qRT-PCR, and co-relation analysis to achieve this. The high point of the presented study is the correlation analysis with clinical factors. The study is nicely written and presented. The following areas of concern need to be addressed.

1) The genes selected for this study are significant in skin homeostasis and diseases. However, the criteria for presenting them as biomarkers need to be more detailed and clearly outlined.

We have included the following text:

According to literature [34-41] involucrin and human β-defensin-2 seemed to be good candidates for differentiation between atopic dermatitis and psoriasis vulgaris as their higher concentrations were described in psoriasis vulgaris. Neither TPP2 nor PSMB8 are recognized biomarkers in differentiating the diagnosis of AD and PV. However, since their participation in connection with these diseases has been described in few works [28,32], we decided to investigate whether these two molecules differ between AD, PV and controls. We decided to rely on the mRNA expression analysis of all proteins to be able to compare their expression at the mRNA level. Involucrin is a major marker of skin cell differentiation [36,39], and human beta defensin -2 is a marker of inflammation [37,40,41]. In turn, for the immunoproteasome, the activity - essential subunit is iB5 (PSMB8) [21,27,42] and for TPP2 - the most conserved protease - we chose the most frequently studied sequence to have a reference for other works [29,43]. Since the size of the sample that can be taken from the patient is limited - it was more reasonable to study the reference molecules at the mRNA level. Additionally, for markers of inflammation and skin differentiation - we studied expression also at the protein level to have a second point of reference.

2) Table 1 and Figure 1 are redundant and almost present the same information.

We have moved  Table 1 content to Supplementary data 1

3) Use ROC curve analysis to present the specificity and sensitivity of these genes in distinguishing healthy skin from AD or PV. 

The ROC curve analysis has been done and included under the Algorithm and we have added the text:

According to the results of the algorithm build with the help of neurological network the relative expression of hBD-2 mRNA and of PSMB8 mRNA turned out to be the most valuable parameters in differentiation between AD, PV and C biopsies. ROC curve analysis of hBD-2mRNA and PASMB8 mRNA confirmed, that hBD-2 mRNA is the best parameter to discriminate inflamed from healthy tissues with the highest sensitivity and specificity. PSMB8 mRNA was more helpful than hBD-2 mRNA in differentiating between AD and PV biopsies, although with similar sensitivity and specificity (Figure 8).

4) Compare available clinical tests for AD and PV with these genes side by side to show the presented parameters are better or equivalent to currently used methods in the clinic. 

In real life AD and PV are recognized by clinical examination. In some patients proper diagnosis is demanding.  Histopathological examination plays only supportive role. There are no available histopathological or other differential tests so far.

We have included the text:

About AD:

Histopathologic picture reveals acute, subacute or chronic inflammation with varying degrees of spongiosis [4]. Total IgE immunoglobulin level may be elevated. Part of AD patients suffer from asthma, allergic rhinitis or allergic conjunctivitis [5].

About PV:

There are two subtypes of skin psoriasis: psoriasis vulgaris and pustular psoriasis. Both may evolve into erythroderma. Depending on the activity of the disease, psoriasis vulgaris may be active guttate, unstable exudative or chronic stable. Lesions can be scattered on the trunk, head and extremities or localized. Although inverted psoriasis has predilection to skin folds, psoriatic lesions usually are placed on the extensor surfaces, unlike most of AD changes. Primary lesions of PV are dermo-epidermal papules covered with silvery white scales [12] Histopathological examination of a fully developed psoriatic papule reveals hyperkeratosis with parakeratosis and clusters of neutrophils in stratum corneum accompanied by hypogranulosis and acanthosis with elongated rete ridges and drawn out, widen blood vessels [13,14].

About both diseases:

Clinically AD may mimic PV, moreover primary diagnosed AD may transform into PV and vice versa There is an observation of coexistence of PV and AD features in the same inflammatory lesions, especially in children. These lesions are called PD (psoriasis dermatitis) or PsEma and with time can evolve into either AD or PV. [17].

Additionally histopathological examination may show both spongiosis and parakeratisis with hypogranulosis that make differential diagnosis impossible [18]. There is no commonly available and clinically approved test for differentiation between biopsies from AD and PV lesions.

5) Itch is a demarcating feature of AD and is often not present in PV. Only a small fraction of PV patients present with itch. How many PV patients show itch as a clinical feature? Could itch be a distinguishing factor between healthy skin, AD, and PV?

Unfortunately, itch sensation is not a good differentiation marker.

We have added Supplementary data 2 with report of itch sensation in our patients and we have included the text:

Itching sensation is present in both studied diseases. Among our patients with inflammatory skin diseases all except one patient with PV reported itch with different intensity (supplementary data 2). There are clinical observations that many, if not the most PV patients, complain about itch of skin lesions. For this reason, itch cannot be used to differentiate between these two diseases.

6) Articles cited to show psoriasis prevalence are old. Use recent articles and data to show psoriasis prevalence. 

We have updated the literature and have written the text:

Prevalence of psoriasis in adults is between 0.14% and 1.99% while in children between 0,02% and 0,22%

Round 2

Reviewer 2 Report

Comments and Suggestions for Authors

The authors provided detailed responses to raised comments and added new results to support their claims. I do not have any further concerns about this study.